# Piezoceramic-Based Damage Monitoring of Concrete Structure for Underwater Blasting

**DOI:** 10.3390/s20061672

**Published:** 2020-03-17

**Authors:** Jianfeng Si, Dongwang Zhong, Wei Xiong

**Affiliations:** 1College of Science, Wuhan University of Science and Technology, Wuhan 430065, China; sijian.feng@163.com (J.S.); xiongweiwust@163.com (W.X.); 2Hubei Province Key Laboratory of Systems Science in Metallurgical Process (Wuhan University of Science and Technology), Wuhan 430081, China; 3Hubei Province Intelligent Blasting Engineering Technology Research Center, Wuhan 430065, China

**Keywords:** underwater blasting, damage reduction, damage monitoring, energy-relieving structure, piezoceramic transducer

## Abstract

This paper developed a piezoelectric-transducer-based damage detection of concrete materials after blasting. Two specimens (with or without an energy-relieving structure) were subjected to a 40 m deep-underwater blasting load in an underwater-explosion vessel, and their damage was detected by a multifunctional piezoelectric-signal-monitoring and -analysis system before and after the explosion. Statistical-data analysis of the piezoelectric signals revealed four zones: crushing, fracture, damage, and safe zones. The signal energy was analyzed and calculated by wavelet-packet analysis, and the blasting-damage index was obtained after the concrete specimen was subjected to the impact load of the underwater explosion. The damage of the two specimens gradually decreased from the blast hole to the bottom of the specimen. The damage index of the specimen with the energy-relieving structure differed for the fracture area and the damage area, and the damage protection of the energy-relieving structure was prominent at the bottom of the specimen. The piezoelectric-transducer-based damage monitoring of concrete materials is sensitive to underwater blasting, and with wavelet-packet-energy analysis, it can be used for postblasting damage detection and the evaluation of concrete materials.

## 1. Introduction

### 1.1. Research Background

Water conservancy, and hydropower, civil, and transportation engineering involve a plethora of infrastructure construction, such as the rock foundations of hydraulic structures (mainly dams and bridge-pier foundations), and various types of construction and excavation platforms [1]. Blasting is currently the most commonly used construction method for large foundation excavation and forming. A large amount of energy released after the explosion not only fragments the rock mass in the excavation area, but also causes irreversible dynamic disturbances to the retained rock mass in the damaged excavation area [2,3]. The integrity and physical mechanics of the rock mass in the damaged excavation zone deteriorate at various degrees [4], and have direct impact on the quality of the foundation construction, the economic benefits of the project, and the overall construction progress. Therefore, controlling excavation damage and reducing blasting influence on the rock-foundation protection layer is a focus of engineering-practice and rock-dynamics research [5].

The main methods to control the damage of rock mass are smooth blasting, presplitting blasting, shallow-hole weak blasting in the protective layer, mechanical crushing, and static crushing [6]. Smooth and presplitting blasting offer excellent results, but construction is complicated and inefficient. Although shallow-hole weak blasting in the protective layer, and mechanical and static crushing are relatively simple, construction efficiency is limited. Therefore, the general contradiction between excavation quality and efficiency is common in excavation construction and blasting foundation. Moreover, smooth and presplitting blasting cannot be used in the excavation of an underwater-foundation platform because of the influence of the drilling direction. Because the secondary positioning of underwater drilling is extremely difficult, shallow-hole weak blasting is also impractical. An alternative blasting construction technology that can obtain better horizontal foundation surface is still lacking for the excavation of underwater foundations.

Excavation shaping and the damage-control technique for a rock foundation involves a shock-relieving structure (consisting of a spherical energy-relieving block with high sonic impedance and a flexible cushion with low sonic impedance) placed at the bottom of a borehole. Stress transmitted to the base protection layer is laterally gathered to the bottom of the blast hole and to the upper rock mass through the reflection effect of the high-sonic-impedance spherical energy-relieving block to strengthen the fragmentation between the upper and adjacent rock mass. Meanwhile, the reflection of the energy-relieving block and the isolation of the lower loose-sand layer are used to reduce the blasting damage of the bedrock mass at the bottom of the hole. The application of this technology in the blasting construction of foundation excavation can strengthen rock fragmentation in the horizontal direction, improve the one-time forming effect of the excavation surface, reduce the scope of blasting damage in the direction of the retaining rock, ensure the quality of foundation excavation, and improve construction efficiency. Figure 1 shows a typical energy-relieving structure that has been widely used in the excavation of dam foundations. However, the application of this method in underwater drilling and blasting has not been reported.

Meanwhile, necessary damage-detection methods are indispensable to determine the damage control of the energy-relieving structure on the foundation during underwater drilling and blasting.

### 1.2. Damage Detection of Rock and Concrete Materials

The most popular nondestructive testing techniques in blasting engineering are the ultrasonic-velocity method [6,7], ground-penetrating-radar (GPR) techniques [8,9], and impact-echo approaches [10]. Natural initial damage (microcracks and holes) in rock mass develops and extends until a crack develops under the blasting load. Reflection, diffraction, and scattering occur when the ultrasonic longitudinal wave reaches these structural interfaces (cracks). Therefore, P-wave velocity decreases due to the extension of the propagation path, and the P-wave reduction degree is associated with crack width and number. Lai et al. [11] adopted the ultrasonic-velocity method to detect the damage of ultrahigh-performance concrete under repeated penetration and explosions at different depths. Zhang et al. [12] found that the damage zone can be relatively easily discriminated using P-wave rise time. Aldas et al. [13] found weak zones and closed or compact fracture zones that need a special type of blast design that can be identified by the GPR method. Fuchs and Keuser [14] proposed an empirical model for reinforced concrete slabs to describe damage after high dynamic loading through the impact-echo method. Tashakori et al. [15,16] monitored the dynamic response of the structure, and proposed implementing the heterodyne structural-health-monitoring (SHM) method.

Although the above nondestructive testing technology is often used in the application of large cracks, caves, or intercalations, damage detection is not suited to microstructural features such as microcracks. Thus, a new detection technology is required to measure the degree of rock damage caused by a blasting load and to verify the energy-relief technique.

In this study, a wave-based method was adopted for health-monitoring purposes. A pair of piezoceramic patches bonded to the outer surface of the specimen were used as the actuator and the sensor for detecting possible blast-induced damage inside the two concrete specimens. Wavelet-packet analysis was used as a signal-processing tool to analyze the sensor signal for health monitoring. Experiment results indicated that the energy-relieving technique is appropriate for underwater-blasting-damage control, and that the SHM method is sensitive to damage induced by underwater blasting.

## 2. Detection Principle 

### 2.1. Piezoelectric Transducers and Active Sensing Based on Piezoelectric Transducer

In recent years, with the increasing demands for structural-damage detection [17,18], structural health monitoring (SHM) has rapidly advanced [19,20], and piezoelectric transducers are widely used in structural health and damage detection [21,22]. They have low cost, broadband-frequency response [23,24,25], energy-harvesting capacity [26,27] and the ability to function as both actuators and sensors [28,29]. Piezoceramic transducers have high piezoelectric constants and electromechanical coupling coefficients, and have good energy-conversion characteristics [30]; they can be used as actuators due to their small dielectric loss, and as sensors owing to their small capacitance and high resistance. The Curie point of some piezoceramic transducers is above 300 °C, which ensures their normal working stability in normal- and relatively high-temperature environments. Therefore, piezoelectric transducers are widely used in daily life and scientific research [31,32].

At present, structural health-detection technology based on piezoelectric transducers can be divided into two categories: passive and active damage-detection technologies. Passive detection is based on the response of the sensor embedded in the structure or bonded on its surface without involving the actuation function. Often, piezoelectric sensors are used to directly measure structural acceleration, strain, and acoustic emission [33,34,35]. The active damage-detection technology uses an actuator to induce the stress wave signal, and a sensor to detect the travelling stress signal. The health status of the structure is evaluated by using an algorithm [36]. Active damage-detection technology using piezoelectric transducers can be further divided into the electromechanical-impedance (EMI) method [37,38] and active-sensing methods [39,40]. The EMI method involves only a single piezoceramic transducer that functions as both an actuator and a sensor [41,42]. Meanwhile, in active-sensing methods, at least one pair of transducers are needed: one functions as an actuator and the other functions as sensor [43,44]. Since active-sensing methods requires much lower sampling frequency than that of the EMI method, they have found many applications, such as bolt-connection [45] and timber monitoring [46]. 

Piezoceramic transducers are generally fragile and not suitable for direct deployment to concrete structures without proper packaging. Song et al. [47,48] developed a piezoelectric-based smart aggregate that offers proper protection to piezoceramic-patch transducers, and the protected transducer can be embedded in concrete structures for multifunctional monitoring. In addition, an associated damage index based on wavelet-packet analysis was developed to help interpret the data. The development of smart-aggregate technology accelerated research on piezoceramic transducers in concrete structures in many applications, such as hydration monitoring [49,50,51], impact detection [52,53,54,55], blast-damage monitoring [56], bond-slip or debonding detection [57,58,59], and concrete compactness monitoring [60,61]. 

On the basis of the successful deployment of piezoceramic-based active-sensing technology, here we developed a new method for damage-monitoring concrete specimens with and without the energy-relieving device before and after blasting. Compared with the sensor embedded in the structure, we first used the surface-sticking method to measure blasting damage. This experimental approach proves the reliability of the method; it is economical and convenient, and sensors can be reused.

### 2.2. Piezoceramic Transducers

Among various piezoceramic materials, lead zirconate titanate (PZT) has a strong piezoelectric effect, is commercially available, and was adopted in this research to develop the piezoceramic transducers. As shown in Figure 2, a PZT disk was encapsulated by a metal shell with an outer diameter of 24 mm and a height of 10 mm. The amplification circuit, which was separated by the magnetic steel layer, was also integrated with this transducer using bisphenol A epoxy resin when it was used as a sensor. The transducer was connected to a shielded cable and connected to the data-acquisition instrument or a signal-generating instrument through the Bayonet Nut Connector(BNC) connector.

### 2.3. Piezoceramic-Enabled Active Sensing

Lile any other piezoelectric material, PZT has a direct piezoelectric effect and an inverse piezoelectric effect. PZT transducers can be embedded in or bonded on a concrete structure for health monitoring [62,63,64]. In this research, a PZT-based active-sensing method is proposed to monitor blast-induced damage in a concrete structure; Figure 3 illustrates this principle. There were two pairs of the PZT transducers bonded on the outer surface of the specimen. One pair was placed above the explosion point, which was at the center of the specimen, and another pair was placed below the explosion point and cushion block. It was expected that the top part of the concrete specimen would experience damage due to the blast. The bottom part of the specimen would experience less damage due to the cushion block that absorbs blast energy. For the top pair of PZT transducers, under an excitation signal, the PZT actuator generated a stress wave due to the inverse piezoelectric effect, and the stress wave propagated in the concrete specimen. The PZT sensor transformed the measured stress wave into electrical-signal output due to the direct piezoelectric effect. Because of damage in the structure, the amplitude, energy, propagation time, mode, and waveform of the signal changed. By comparing the signals to those of the healthy state of the structure, we could determine whether the structure was damaged, and with further analysis, the location and degree of the damage. In the upper part of the specimen, due to the existence of cracks in the specimen, the amplitude and energy of the piezoelectric signal received on the right side were smaller than those of the bottom part under the same signal excitation on the left side.

### 2.4. Proposed Blasting-Damage Index

Wavelet-packet and discrete-wavelet transform are commonly used signal-analysis methods. In the discrete-wavelet-transform method, the signal is decomposed into an approximate low-frequency part and a detailed high-frequency part. Thereafter, the low-frequency detail is decomposed again, whereas the high-frequency part is not decomposed any further. Contrarily, wavelet-packet transform contains high- and low-frequency components when decomposing the signal, and this method is less affected by noise. When there is a change in the structure state, the energy of each frequency band of the response signal also changes after wavelet-packet transform; therefore, the wavelet-packet-energy spectrum can be used to identify structural damage. Cheraghi et al. [65] used wavelet packets to identify the damage in a pipeline structure; they employed Fourier, wavelet, and wavelet-packet transform to evaluate pipeline damage. Their results indicated that wavelet-packet transform yields better recognition accuracy. Peng, and Hao et al. [66,67] proposed the average wavelet-packet-energy change-rate damage-identification index to identify the implantation situation of submarine oil and gas pipelines. Numerical analyses and experiments proved the effectiveness of the method, which has high accuracy and noise immunity. These studies showed that the wavelet-packet-energy spectrum is a significantly effective method for the identification of structural damage.

In this experiment, several irregular small cracks were generated inside the specimen after blasting. When the transmitted signal passed through these cracks, it resulted in refraction, transmission, and other effects; hence, the complexity of the frequency component of the signal increased. On the basis of characteristics of the complex frequency components of this test signal, small cracks were found to be more sensitive to high-frequency component signals, and wavelet-packet energy could more evenly divide the noise into various frequency bands with good noise resistance. Therefore, to measure the damage of concrete specimens with or without an energy-relieving structure, wavelet-packet decomposition was selected as the signal-processing tool.

On the basis of wavelet-packet analysis, a blasting-damage index that represented the loss of transmitted energy due to blasting damage was established.

Signal X represents the detected signal. It was first filtered using a Butterworth filter, and the $2^{n}$ signal set $\left\{x_{1},x_{2},\ldots x_{j},x_{2^{n}}\right\}$ was then obtained via n-level wavelet-packet decomposition. $x_{j}$ is the decomposed signal, which could be expressed as
(1)$$x_{j}=\left[x_{j,1},x_{j,2},\ldots ,x_{j,m}\right]$$
where *m* is the number of the sampled data, and *j* is the number of the frequency bands (j = 1,2,...$2^{n}$). Energy $E_{j}$ of each decomposed signal $x_{j}$ is defined as
(2)$$E_{j}=x_{j,1}^{2}+x_{j,2}^{2}+\ldots +x_{j,m}^{2}$$

Root-mean-squared deviation (RMSD) was used to represent the damage index of concrete materials, that is, the damage index was established by computing the RMSD of the energy before and after blasting. Thus, the *i*th damage index can be defined as
(3)$$\eta =\sqrt{\frac{\sum_{j=1}^{2^{n}}{\left(E_{i,j}-E_{h,j}\right)}^{2}}{\sum_{j=1}^{2^{n}}{\left(E_{h,j}\right)}^{2}}}$$
where $E_{h,j}$ is the energy of the decomposed sensor signal at the *j*th frequency before blasting, and $\;E_{i,j}$ is the energy of the signal measured in the *j*th frequency band at the *i*th time after blasting. In this work, only two tests were performed before and after blasting, *h* = 1 and *i* = 2, where 0 ≤ η ≤ 1. When η = 0, the material/structure remains undamaged; when η = 1, a through crack occurs, and $E_{i,j}\to 0$ at this time.

## 3. Experiment Setup and Procedure

### 3.1. Specimens

The comparative method was used to test the effect of the energy-relieving structure. Two C30 concrete cubes with 245 × 245 × 300 mm dimensions were prefabricated, as shown in Figure 4. Steel bars with a diameter of 12 mm and a length of 100 mm were embedded at the center of the top surface perpendicular to the base surface. The steel bars were removed after the specimens were cured for 12 hours, and 12 × 100 mm blast holes were formed after the specimens were completely hardened (after curing for 28 d). Six standard samples (100 × 10 × 100 mm) were poured to test the compressive and tensile strength of concrete. The basic mechanical parameters of C30 are listed in Table 1.

### 3.2. Experiment Setup

The multifunctional piezoelectric signal-detection and -analysis system is shown in Figure 5. This system had two transmitting channels and eight receiving channels. The transmitting signals were divided into pulse signals and analog signals. The amplitude, frequency, and time-holding of the transmitted signal could be freely adjusted. The transmission signal was a single pulse signal with a pulse amplitude of 100 V, pulse width of 15 μs, encoding bit of 2, duration of 30 μs, and sampling rate of 2 MHz, as shown in Figure 6.

To simulate underwater drilling blasting, we used the horizontal water-medium explosion vessel, as shown in Figure 7. The explosion vessel was 2 m in diameter, 2 m in length, and 0.5 m at both ends. Two 280 mm (diameter) sight windows in opposite directions, one a manhole with a diameter of 500 mm, and one a detonation interface with a diameter of 300 mm, two cable outlets with a diameter of 280 mm, one static pressure test port, electric pressure, and pressure-relieving ports, electric water inlet and outlet ports, were set in this explosion vessel. One pressure test pump was connected to the pressure port with a power of 2 m³/min. The explosion vessel could simulate an explosion of 10 g TNT in 200 m depth (with pressure of up to 2 MPa). The simulated depth used was 40 m (0.4 MPa).

In Specimen F5, the blast hole was filled with coarse sand to 70 mm, and a steel ball with a diameter of 10 mm was the placed inside. Then, the charge consisting of one No. 8 steel detonator and 2 g pentaerythritol tetranitrate (PETN) was added, and the blast hole was blocked with soft blast mud. In F6, the blast hole was first filled with fine sand to 70 mm; then, the charge consisting of one No. 8 steel detonator and 2g PETN was filled, and the blast hole was plugged with soft clay. The specific scheme is summarized in Table 2.

### 3.3. Experiment Procedure

The initial states of Specimens F5 and F6 were measured by the wave-based method before blasting. As shown in Figure 8, four points (1–4) were taken in the horizontal direction, and five points (A–E) in the vertical direction with column and row spacing of 49 and 50 mm, respectively; a total of 20 points were tested. These points were determined in advance and marked using paint. During the test, the actuators and sensors were affixed on the measuring points using Vaseline.

The two specimens were filled according to the experiment plan shown in Table 2, with the energy-relieving ball, charge, and stemming installed. The specimen was placed in the water-medium explosion vessel, which was sealed and filled with water, as shown in Figure 9. Then, the pressure-valve switch was turned on, and static pressure was adjusted to reach 0.4 MPa. The pressure pump was later stopped, and the pressure valve was closed. The detonation line was connected to the detonator, and the system was detonated. Finally, pressure was released, water was drained, and the sample was removed. The specimen was placed in aeration for 24 hours, and piezoelectric data were obtained after blasting. The entire experiment procedure is shown in Figure 10.

## 4. Experiment Results and Analysis

As shown in Figure 11, cracks appeared on the upper surface of F5 (Figure 11a) and F6 (Figure 11b) when the specimens were taken out after blasting. Both specimens developed cracks at the upper section, whereas the bottom of the specimens withstood the pressure. Upper surface crack and crack width in F5 with the energy-relieving structure were more uniform than those in F6, and the number and length of side cracks in F5 were less than those in F6, as shown in Figure 11c,d.

Measured-waveform-based active sensing was performed before and after blasting, as shown in Figure 12a–f. The amplitude of the wave became larger from the top to the bottom of the blast hole and the bottom of the specimen, respectively, indicating fewer internal cracks in the concrete cube. F5 had a certain amplitude waveform at C2, while it was still a horizontal line for F6 at C2 after blasting. The damage depth of F6 was larger than that of F5, demonstrating that the energy-relieving structure protected against rock damage at the bottom of the specimen.

The peak value of piezoelectric signals at various measurement points of Specimens F5 and F6 before and after blasting are shown in Figure 13a,b. The entire curve could be divided into four sections according to the position of the blast hole and the graph of voltage-signal peak: The first section was where the blast hole was located, i.e., the area 8 cm below the upper surface of the specimen, where the cracks were obvious after blasting. The signal peak substantially lay on the same horizontal line after blasting; this showed that most of the energy was absorbed due to the cracks, and only a fraction of the energy was sensed. This section could be defined as the crush zone. The second section is the area defined as the fracture zone (8–15 cm below the blast hole), where a few closed fractures still took place. The peak value of the waveform fluctuated after blasting. Following the fracture zone was the zone where the surface of the specimen was crackfree, and the peak (15–20 cm below the hole) was similar to that of the fracture zone, meaning that energy loss was minimal; this area was defined as the damage zone. The 10 cm deep region at the bottom of the specimen was defined as the safe zone, where the peak value of the waveform remained unchanged after blasting, indicating that this area was not affected.

The preblasting peak value of piezoelectric signals in Zones I and II of F5 and F6 was greater than the postblasting peak value, and Zone I was more obvious than Zone II. The peak value of the piezoelectric signal fluctuated within a certain range before and after blasting in Zone III and became closer in Zone IV. Damage variation decreased from the upper to the lower surface of the concrete and was consistent with the general recognition law.

Subsequently, according to the damage index established on the basis of wavelet-packet analysis as described in Section 2.4, the damage index of Specimens F5 and F6 corresponding to the four zones was analyzed. First, wavelet-packet decomposition was applied to the test signals before and after the explosion of each measuring point. When choosing the wavelet base, no related rigorous theoretical basis was found. The db series is a relatively common orthogonal wavelet base; therefore, we used it from the initial stage of this study. After consulting numerous articles, we found that db2, db5, and db9 were all available for damage-signal analysis. Therefore, we used the three aforementioned wavelet bases for analysis, and the effects were not substantially different. Considering the amount of calculation, we used db2 for analysis. After analysis, we found that the result of damage analysis was in good agreement with our expected results, which further justified our choice of the db2 wavelet base. The selection of the wavelet-packet decomposition series is related to the analyzed signal, and there is no unified theoretical basis in the field of damage analysis. We initially used a five-level decomposition trial and then increased the value of n for comparative analyses. According to the results after decomposition, increasing n did not significantly affect the analysis results. Therefore, we chose n = 5. A single measurement point could obtain 32 subsignals after wavelet-packet decomposition, before or after the explosion.

The energy of each subsignal could be obtained using Equation (2). Thereafter, according to Equation (3), the η values of F5 and F6 were calculated, and the average values for Zones I–IV were calculated. The histogram of the damage index of each zone is shown in Figure 14.

The damage indices of F5 and F6 had abrupt changes moving from Zone II to Zone III. The damage index of F5 in Zone III was only 0.281, whereas the damage index of F6 was 0.4812. The average damage index for F5 remained unchanged between Zones III and IV, but it dropped from 0.4812 to 0.2639 in Zone III for F6. This shows that no crack occurred in the damage zone for F5, but F6 had a crackfree height of only 5–10 cm. We concluded that the energy-relieving structure offers better protection.

## 5. Conclusion

With the aid of a water-medium explosion vessel, two specimens with and without an energy-relieving structure were exploded with the same charge under a simulated water depth of 40 m. Active-sensing technology based on a piezoelectric transducer was used to detect the structural health of the specimens before and after blasting. The damage rules and damage index of the two specimens were obtained by statistical comparative analysis of the wave-peak value and wavelet-packet-energy analysis. The following conclusions could be drawn:piezoelectric-transducer-based active-sensing technology can effectively identify the blasting damage of concrete materials with sensitivity to the degree and depth of damage;piezoelectric-signal analysis before and after blasting showed that an energy-relieving structure protects the rock mass below the blast hole in underwater drilling and blasting; andprotective energy-relieving structures can be used in underwater-foundation excavation engineering to mitigate the effects of blasting damage on the surface of the underwater foundation.

## Figures and Tables

**Figure 1 sensors-20-01672-f001:**
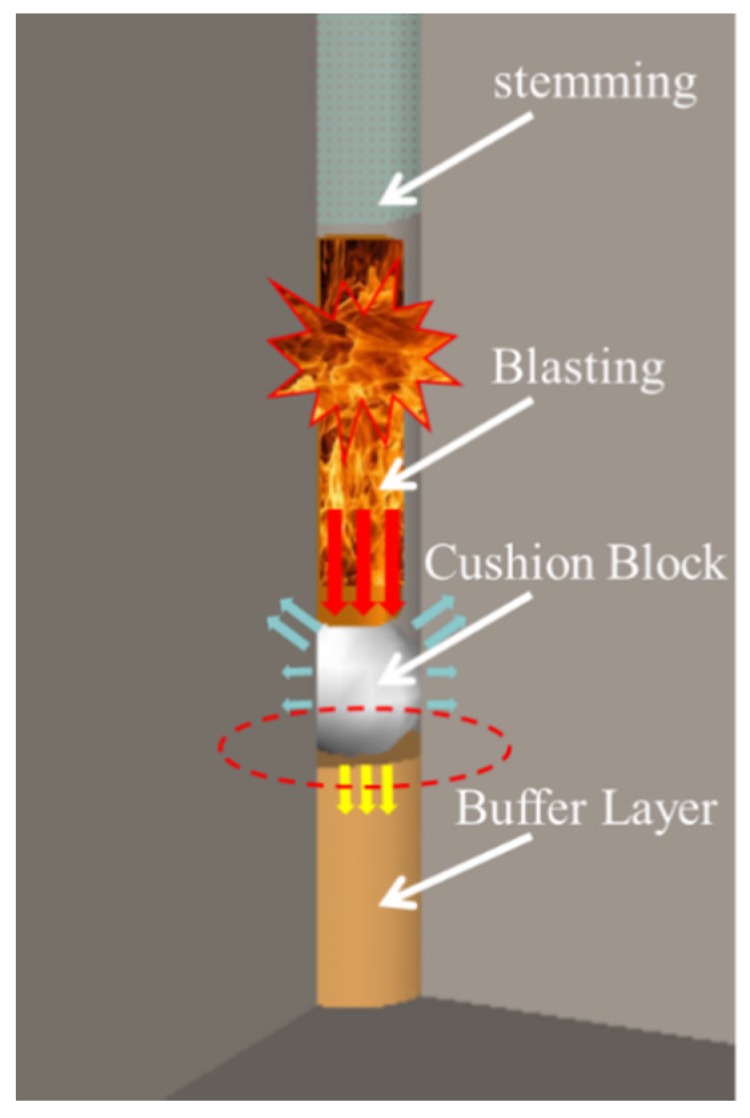
Energy-relieving structure.

**Figure 2 sensors-20-01672-f002:**
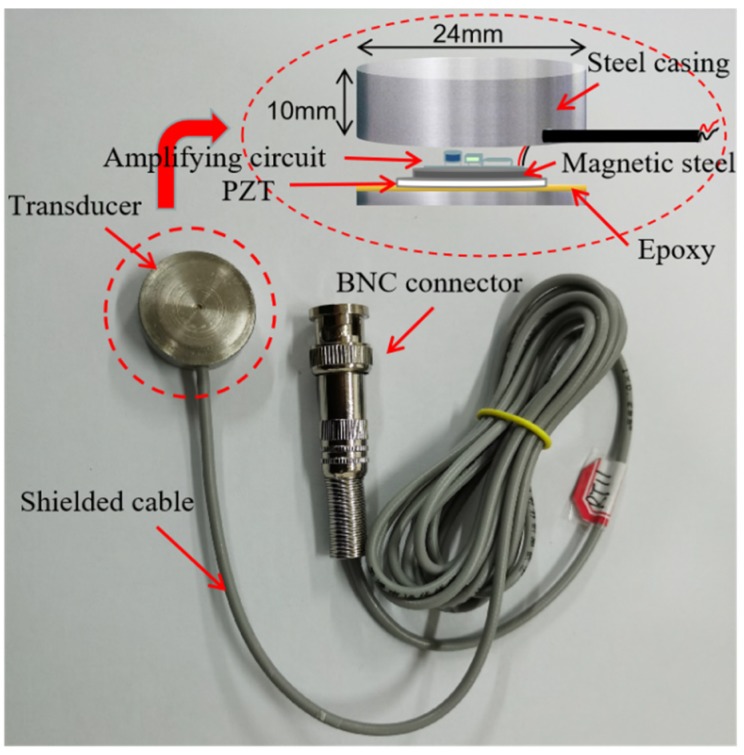
Basic structure of a piezoelectric transducer.

**Figure 3 sensors-20-01672-f003:**
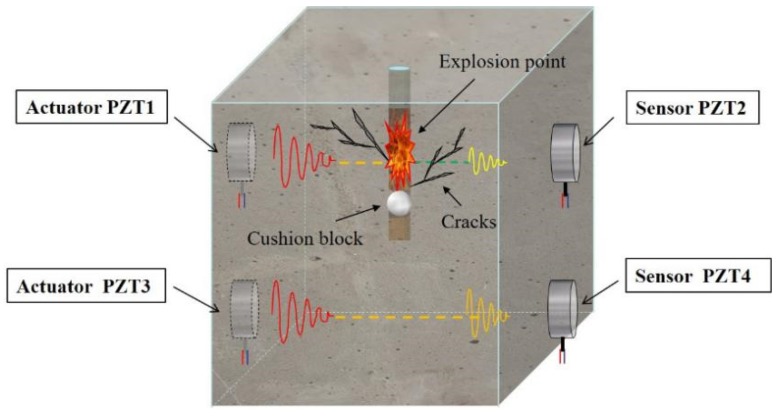
Lead zirconate titanate (PZT)-based active sensing to detect concrete damage.

**Figure 4 sensors-20-01672-f004:**
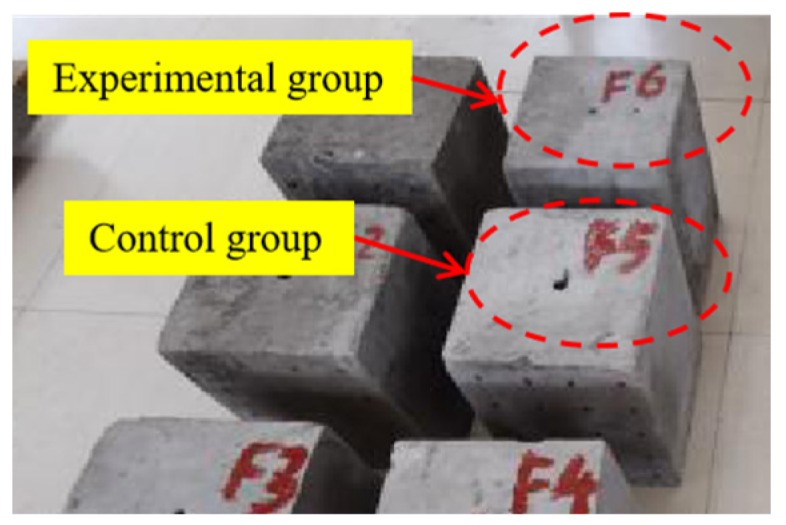
Concrete-cube specimens.

**Figure 5 sensors-20-01672-f005:**
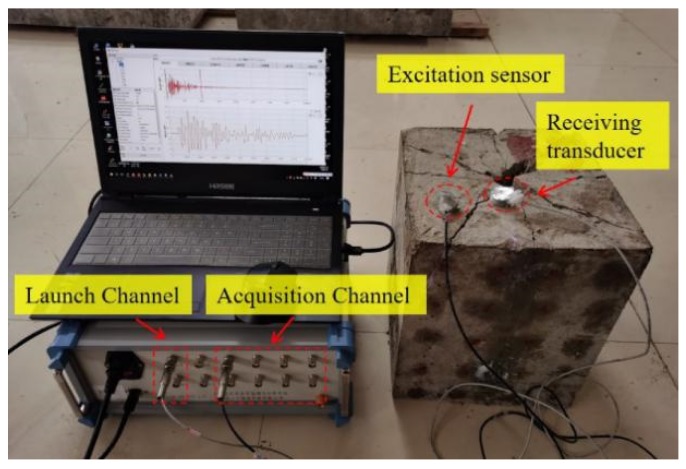
Multifunctional piezoelectric signal-detection and -analysis system.

**Figure 6 sensors-20-01672-f006:**
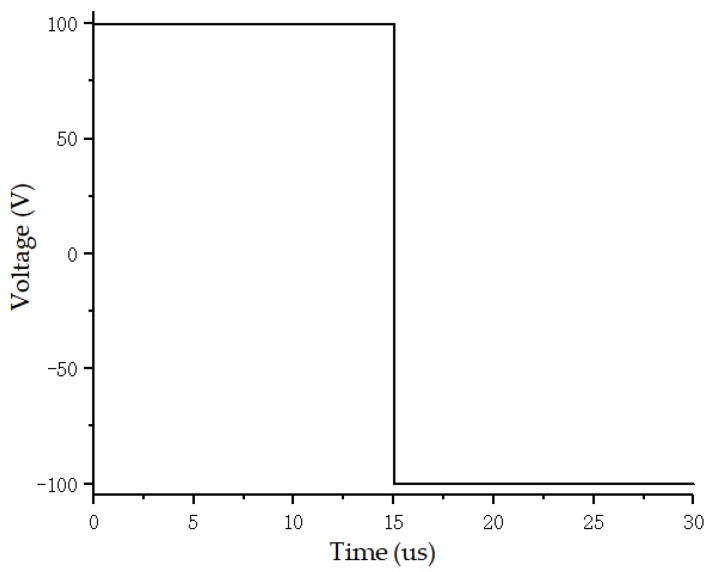
Signal applied to actuator.

**Figure 7 sensors-20-01672-f007:**
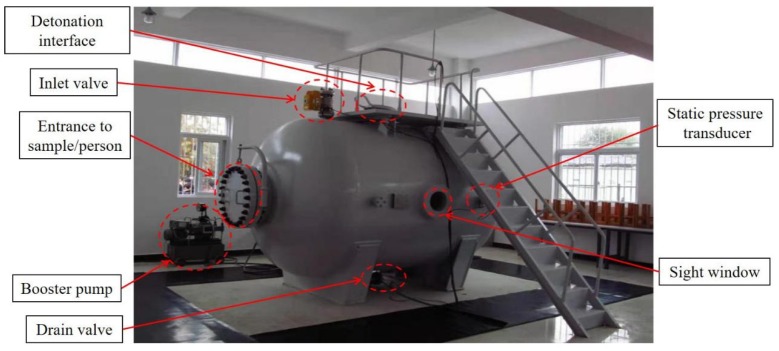
Water-medium explosion vessel.

**Figure 8 sensors-20-01672-f008:**
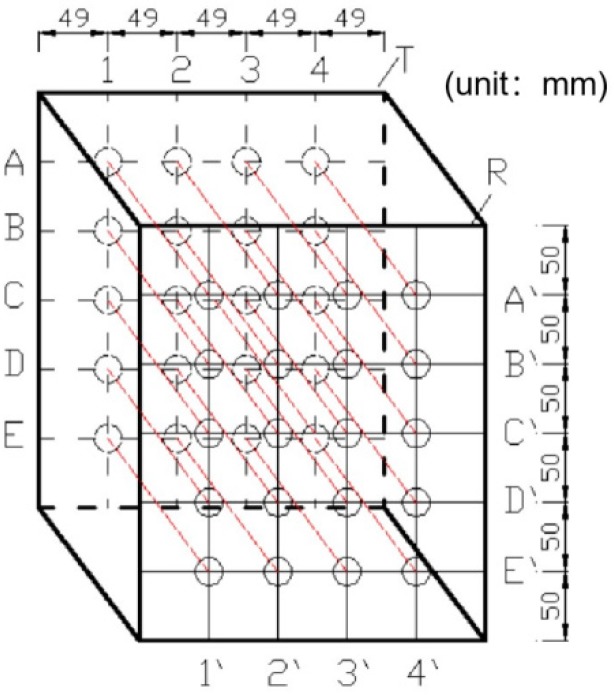
Measurement points.

**Figure 9 sensors-20-01672-f009:**
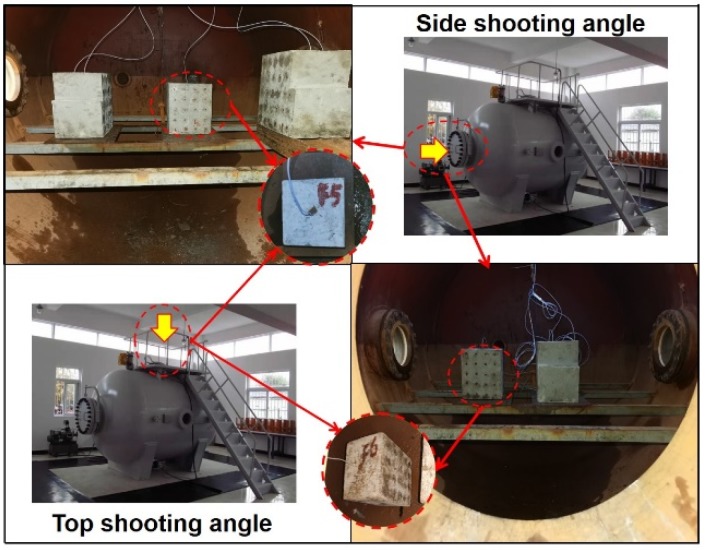
Specimen-placement photographs inside vessel.

**Figure 10 sensors-20-01672-f010:**
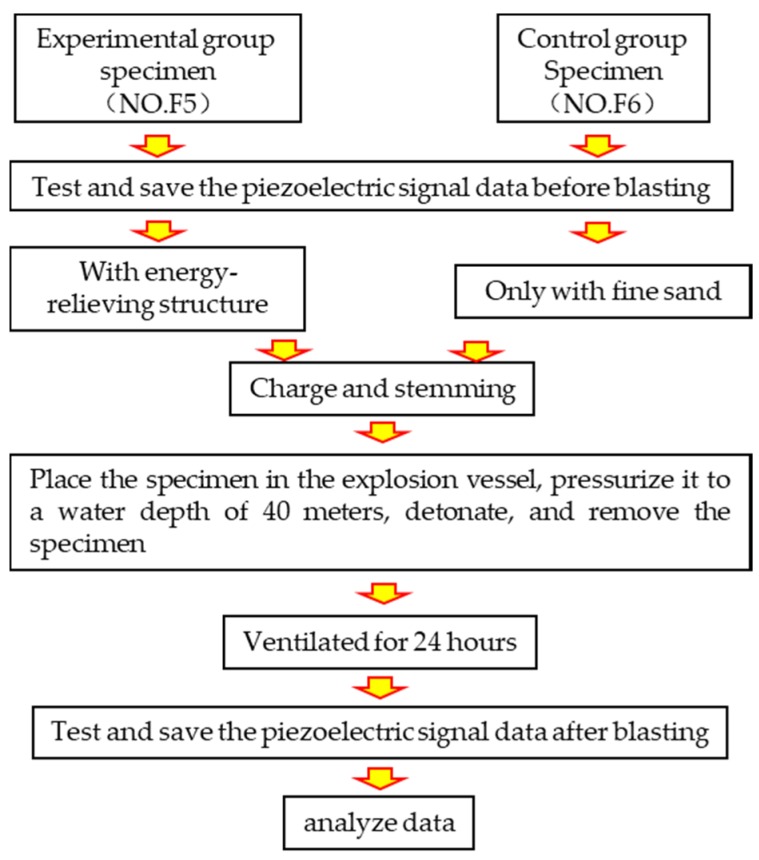
Experiment procedure.

**Figure 11 sensors-20-01672-f011:**
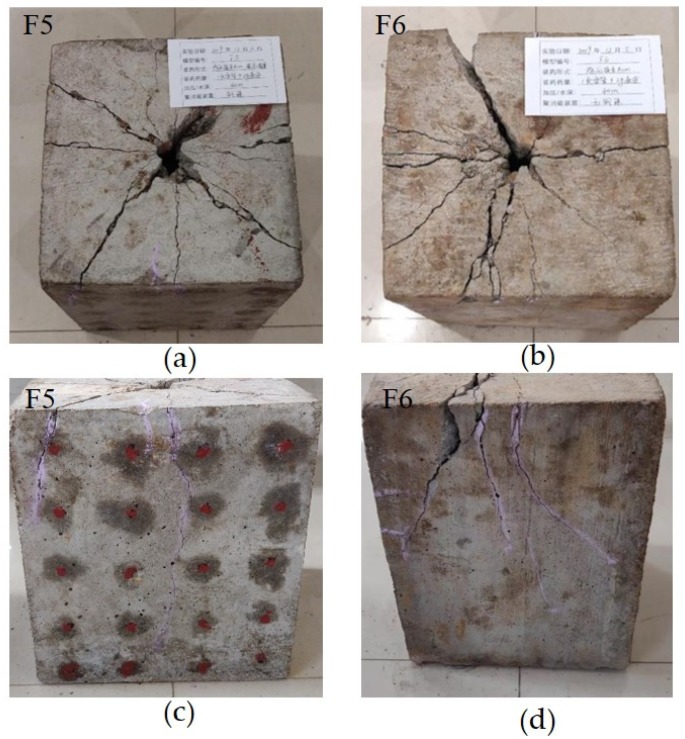
Specimens (**a**,**c**) F5 and (**b**,**d**) F6 after blasting.

**Figure 12 sensors-20-01672-f012:**
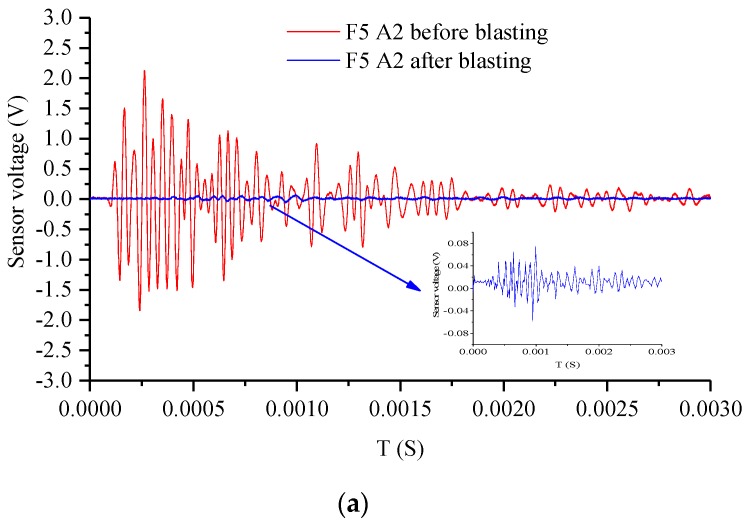
Waveform based on active sensing: test points (**a**) A2 of F5, (**b**) C2 of F5, (**c**) E2 of F5, (**d**) A2 of F6, (**e**) C2 of F6, and (**f**) E2 of F6.

**Figure 13 sensors-20-01672-f013:**
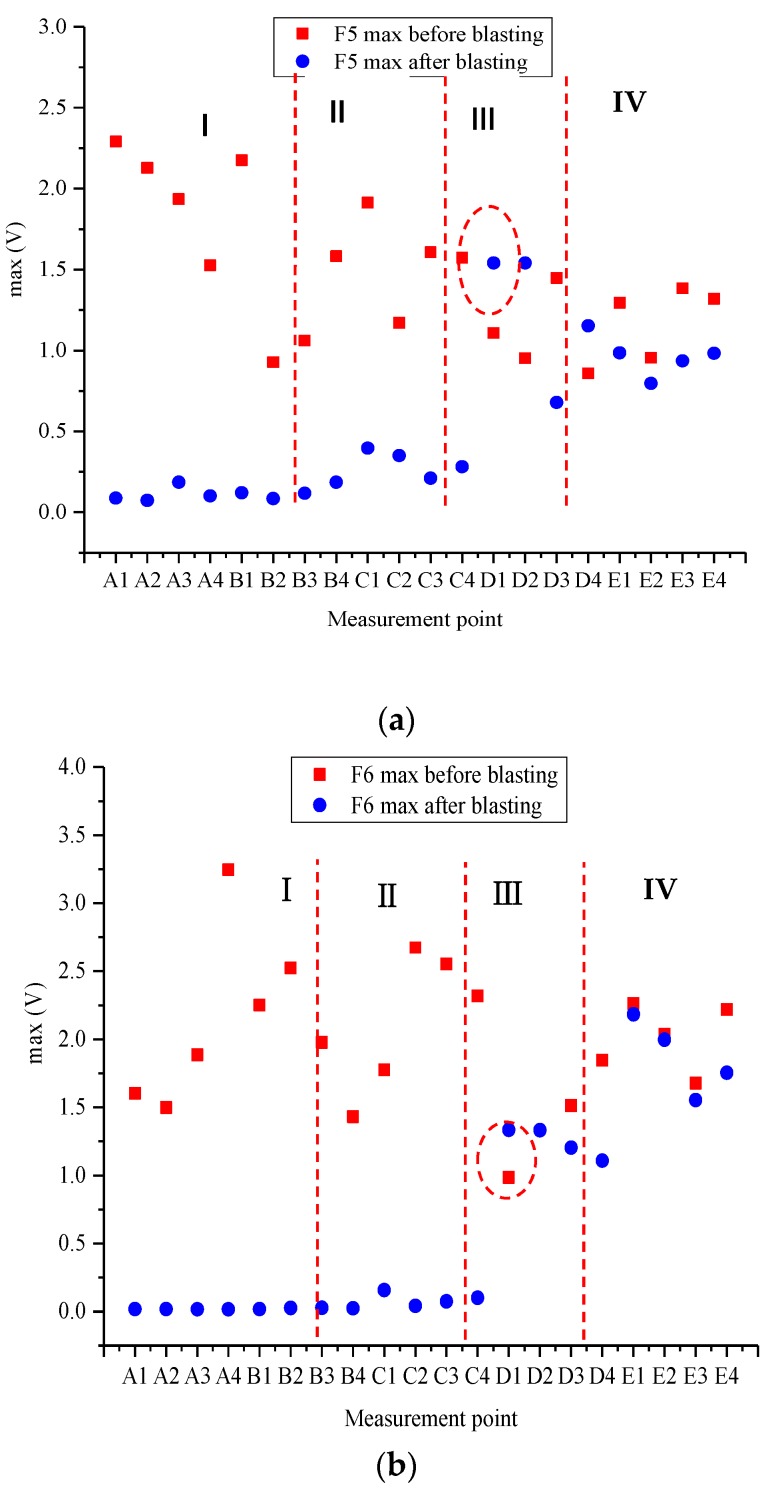
Peak value of piezoelectric signals at various measurement points for specimens (**a**) F5 and (**b**) F6.

**Figure 14 sensors-20-01672-f014:**
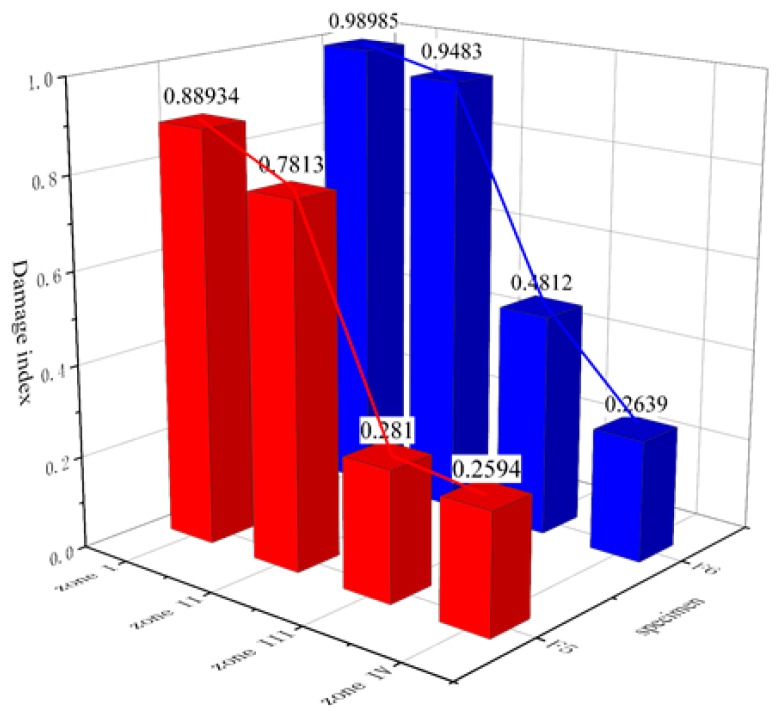
Average damage index of each zone.

**Table 1 sensors-20-01672-t001:** Mechanical parameters of specimens.

	Density (g/cm^3^)	Tensile Strength (MPa)	Compressive Strength (MPa)	Elastic Modulus (GPa)
Concrete C30	2.36	1.87	21.63	27

**Table 2 sensors-20-01672-t002:** Specific scheme.

Group	Sample No.	Charge-Structure Diagram	Charge	Water Depth(m)
Experiment specimen	F5	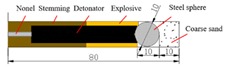	1 detonator + 2 g PETN	40
Control specimen	F6	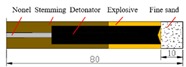	1 detonator + 2 g PETN	40

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
