# Peer review of "Piezoceramic-Based Damage Monitoring of Concrete Structure for Underwater Blasting"

_sensors, 2020, doi:10.3390/s20061672_

Round 1

Reviewer 1 Report

The manuscript presents experimental studies on damage monitoring of concrete structures subjected to underwater blasting using PZT sensors. The authors presented broad literature survey on blasting as well as using of PZTs during structural diagnostics and monitoring. Although, the literature survey is fine, the authors didn’t emphasized the novelty of their study. This novelty should be presented in the light of cited previous studies, e.g. cited studies of Song et al. The paper needs substantial revision before publication. Below, additional detailed comments are presented.

1) In line 191 the authors used the ‘MHz/s’ unit. In this case the sampling rate would be ‘1/s^2’, which is not correct. The unit should be just ‘MHz’.

2) I suggest disconnect the points on plots in Figure 12. The used approximation function is not known, since the authors didn’t test the locations between the presented measurement points. Thus, I suggest to leave datapoints only. The text describing this results below Figure 12 should be appropriately modified.

3) The authors used wavelet packet transform (WPT) as the post-processing technique, however, there is lack of numerous parameters and information. First of all, what was the reason of using WPT instead of discrete wavelet transform (DWT)? Which advantages of WPT were used by the authors in this particular case?

4) If the authors consider all components after wavelet packet decomposition, the determination of energy according to formula (2) gives the full energy of this signal, thus WPT is not necessary to obtain E_j.

5) The authors used ‘n-level WPT’. It is necessary to provide information on what is the value of n and justify the selection of this value.

6) The wavelet and scaling functions used to construct packets are of crucial importance to obtain desired results. It is necessary to provide information on which scaling/wavelet function was used to construct packets and justify their selection.

7) It is not clear why Butterworth filter was used in the pre-processing step. Please explain the necessity of filtering and justify the selection of this particular filter. It would be useful to show it graphically.

8) It would be important to add a discussion to the introductory parts about application of wavelets in damage identification, discuss the advantages and disadvantages of various wavelet transforms in damage identification, and justify the selection of WPT for this study.

Reviewer 2 Report

In this paper, piezoceramic-based damage monitoring of concrete structure for underwater blasting is studied. The manuscript offers innovative results. Hence, the paper is recommended for publication after the following questions were addressed.

  1. In order to clarify the importance of the study, more literature review regarding damage detection, structural health monitoring and non destroctive testing using PZT transdusers should be added to the introduction section. Following papers can be reviewed in this regard:

  • Tashakori, A. Baghalian, M. Unal et al., “Contact and noncontact approaches in load monitoring applications using surface response to excitation method,” Measurement, vol. 89, pp. 197–203, 2016.
  • Tashakori, A. Baghalian, V.Y. Senyurek, M. Unal, D. McDaniel, I.N. Tansel, Implementation of heterodyning effect for monitoring the health of adhesively bonded and fastened composite joints, Appl. Ocean Res. 72 (Mar. 2018) 51–59.

  1. Which type of the epoxy was used to attach the PZTs on the surfaces?
  2. It would be better to add the plots of the signal applied to the actuators.

Reviewer 3 Report

In this paper, piezoelectric transducer-based active sensing was used to detect blasting damage in a simulated underwater environment. A protective energy-relieving structure was used to reduce the blasting damage. The experiment was well designed and conducted. It is a meaningful research. However, the following aspects may need to addressed before the paper can be accepted for publication.

  1. Clearly address the research contribution in the Introduction.
  2. In line 136, authors claimed that "we develop a new method for damage monitoring of concrete specimens...". Why is it new? 
  3. Wavelet packet analysis method described from line 289 to line 312 should be in Section 2, Principle of detection.
  4. How to attach the PZT actuators and sensors to the concrete specimens? How to select the positions?
  5. Proofreading the paper to fix typos and grammatical errors. e.g., 40m-deep=underwater, platforms 1, etc.
  6. Check the citation format. e.g. [30].

Round 2

Reviewer 1 Report

The authors considerably improved the manuscript according the suggestions, however, several questions remain unexplained. The paper is worth publication, but several minor explanations are necessary.

It would be useful to explain why Daubechies wavelet of order 2 was selected. It is also important to note that this wavelet was used as the basis for construction of wavelet packets, since in WPT a set of wavelet packets should be constructed for decomposition both low- and high-frequency components.

After introducing these explanations the paper will be ready to be published.

Reviewer 3 Report

In this revision, authors has carefully addressed my comments and concerns. The paper can be accepted for publication.

Author Response

Thank you very much for your guidance!